# Splicing QTL analysis focusing on coding sequences reveals mechanisms for disease susceptibility loci

Kensuke Yamaguchi [1,2,3,4], Kazuyoshi Ishigaki [5], Akari Suzuki[4],
Yumi Tsuchida [3], Haruka Tsuchiya[3], Shuji Sumitomo[3], Yasuo Nagafuchi [3,6],
Fuyuki Miya[7], Tatsuhiko Tsunoda [8,9,10], Hirofumi Shoda[3], Keishi Fujio[3],
Kazuhiko Yamamoto[4] & Yuta Kochi [2,4] ✉

Splicing quantitative trait loci (sQTLs) are one of the major causal mechanisms in genome-wide association study (GWAS) loci, but their role in disease pathogenesis is poorly understood. One reason is the complexity of alternative splicing events producing many unknown isoforms. Here, we propose two approaches, namely integration and selection, for this complexity by focusing on protein-structure of isoforms. First, we integrate isoforms with the same coding sequence (CDS) and identify 369-601 integrated-isoform ratio QTLs ($i^2$-rQTLs), which altered protein-structure, in six immune subsets. Second, we select CDS incomplete isoforms annotated in GENCODE and identify 175-337 isoform-ratio QTL (i-rQTL). By comprehensive long-read capture RNA-sequencing among these incomplete isoforms, we reveal 29 full-length isoforms with unannotated CDSs associated with GWAS traits. Furthermore, we show that disease-causal sQTL genes can be identified by evaluating their trans-eQTL effects. Our approaches highlight the understudied role of protein-altering sQTLs and are broadly applicable to other tissues and diseases.

Genome-wide association studies (GWAS) have identified thousands of susceptibility loci for complex traits such as autoimmune diseases, metabolic diseases, and cancers; however, the causal mechanisms are not yet fully understood. Although expression quantitative trait loci (eQTL) have been found to be enriched among GWAS loci in disease-relevant cells and tissues[1–3], recent studies suggest that mechanisms other than eQTL explain a substantial proportion of disease heritability[4]. Splicing quantitative trait loci (sQTL) where genetic variants affect alternative splicing[5–11] are a strong candidate mechanism. In the GTEx project, 23% of GWAS loci co-localized with sQTL, while 43% co-localized with eQTL[11]. In addition, genomic loci previously defined as eQTL may share sQTL signals because alternative splicing also affects the gene expression levels. Indeed, intense evaluation of GWAS loci, where eQTL signal coexisted, demonstrated that alternative splicing causes disease is the responsible mechanisms for diseases[12–16].

[1]Biomedical Engineering Research Innovation Center, Institute of Biomaterials and Bioengineering, Tokyo Medical and Dental University, Tokyo, Japan. [2]Genomic Function and Diversity, Medical Research Institute, Tokyo Medical and Dental University, Tokyo, Japan. [3]Department of Allergy and Rheumatology, Graduate School of Medicine, The University of Tokyo, Tokyo, Japan. [4]Laboratory for Autoimmune Diseases, RIKEN Center for Integrative Medical Sciences, Okohama City, Kanagawa 230-0045, Japan. [5]Laboratory for Human Immunogenetics, RIKEN Center for Integrative Medical Sciences, Okohama City, Kanagawa 230-0045, Japan. [6]Department of Functional Genomics and Immunological Diseases, Graduate School of Medicine, The University of Tokyo, Tokyo, Japan. [7]Center for Medical Genetics, Keio University School of Medicine, Tokyo, Japan. [8]Laboratory for Medical Science Mathematics, Department of Biological Sciences, School of Science, The University of Tokyo, Tokyo, Japan. [9]Laboratory for Medical Science Mathematics, Department of Computational Biology and Medical Sciences, Graduate School of Frontier Sciences, The University of Tokyo, Tokyo, Japan. [10]Laboratory for Medical Science Mathematics, RIKEN Center for Integrative Medical Sciences, Okohama City, Kanagawa 230-0045, Japan. ✉e-mail: y-kochi.gfd@mri.tmd.ac.jp

The methodological approaches to identify sQTL can be classified into two types: those based on isoform expression levels[5–7,17,18] and those based on junction read counts[19,20]. A typical example of the former is isoform ratio QTL (i-rQTL) analysis, or transcript ratio QTL analysis, which focuses on the ratio of isoform expression in the gene quantified by the transcriptome assembler. i-rQTL analysis enables a direct understanding of which isoform expression is altered, that is, what kind of protein-structure change occurs. Its disadvantage is that the quantification of isoform expression may be inaccurate, particularly when the isoform annotation is incomplete. This flaw can be overcome by subsequent methods based on raw junction reads without estimating isoform expression. However, when the same junction is shared by multiple isoforms (53.0% of junctions in GENCODE v35), it is difficult to know which isoform expression is altered.

Although previous sQTL analyses revealed that the largest proportion of sQTL changed the untranslated regions (UTR) of isoforms[7,11], which might affect the stability of RNA or translational efficacy, some sQTL changed coding sequences (CDS) by skipping or introducing coding exons. This may substantially impact protein-structure and its function by excluding or including functional domains. In this study, we propose two i-rQTL analyses focusing on CDS. The first is integrated-isoform ratio QTL analysis (i²-rQTL analysis), which integrates isoforms with the same CDS for detecting protein-structure change. The second approach examined sQTL effects on CDS incomplete isoforms, namely GENCODE isoform annotations that contain incomplete but unique CDS fragments. As the latter approach would lead to identification of unknown coding isoforms, we further validated the full-length sequences of disease relevant isoforms by long-read capture RNA-seq. Alternative splicing is a complex event that produces a number of unknown isoforms, and to our knowledge, no sQTL analysis focusing on changes in CDS and confirming the full-length isoforms by utilizing long-read capture RNA-seq has been previously reported. Our approaches show the potential for elucidating the role of protein-altering sQTL in complex traits.

## Results

### Definition of eQTL and sQTL

We previously reported eQTL analysis of six immune subsets (B-cells, CD4+ T-cells, CD8+ T-cells, monocytes, NK cells, and peripheral blood leukocytes (PBL)) from 105 healthy Japanese volunteers[21]. In the present study, we re-analyzed the dataset to further understand the properties of eQTL and sQTL. The term sQTL, in a broad sense, encompasses both QTL for isoform ratios and QTL for ratios of junction read counts. Moreover, the eQTL effect for a specific isoform would influence both the isoform ratio and gene expression level (the sum of isoform expression levels). For the sake of simplicity, we hereafter use the terms gene eQTL, isoform eQTL, and i-rQTL according to the following definitions: 1) genomic loci altering the gene expression level were defined as "gene eQTL", 2) those altering the isoform expression level were defined as "isoform eQTL (i-eQTL)", and 3) those altering the ratio of the isoform expression level were defined as "isoform ratio QTL (i-rQTL)" (Fig. 1a, b). Note that the concept of i-rQTL focuses only on the ratio of the isoform expression

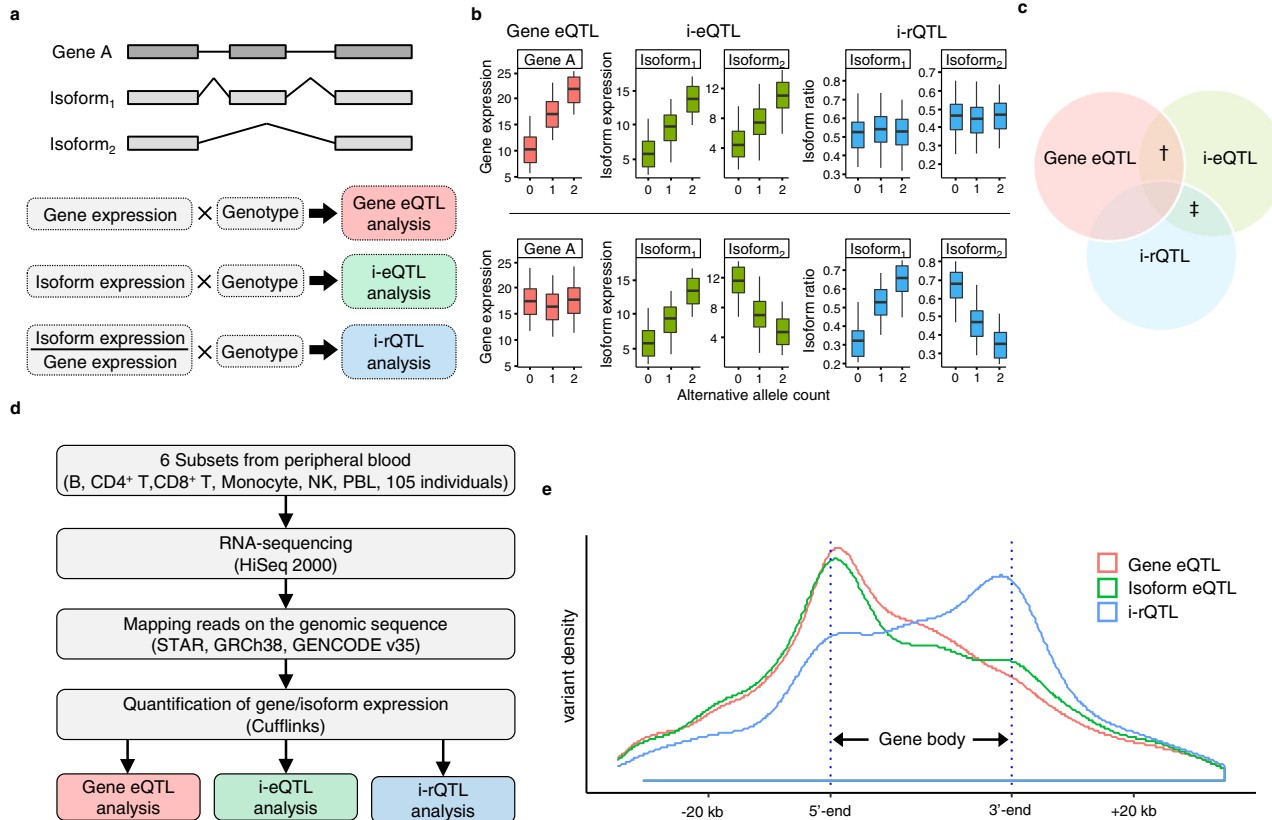

**Fig. 1 | Three types of quantitative trait loci (QTLs); gene eQTL, isoform eQTL and i-rQTL. a** Gene A has 2 isoforms, isoform 1 and isoform 2. A locus affecting the expression level of gene A is defined as gene eQTL. A locus affecting the expression level of isoform 1 and/or 2 is defined as i-eQTL (isoform eQTL). A locus affecting the isoform ratio of isoform 1 and/or 2 is defined as i-rQTL (isoform ratio QTL). **b** An example of gene having both gene-eQTL and i-eQTL effects but not i-rQTL effect (the upper panel) and that having both i-eQTL and i-rQTL effects but not gene-eQTL effect (the lower panel). **c** The three types of QTLs are not independent but overlapping concepts. † and ‡ indicate the genes in the upper and lower panels of **b**, respectively. **d** Data preparation and processing from 6 peripheral blood cells for the QTL analyses. **e** Distributions of top QTL variants relative to the gene body. In the figure, the length of the gene body is 30,000 bp, which is the average length of all isoforms.

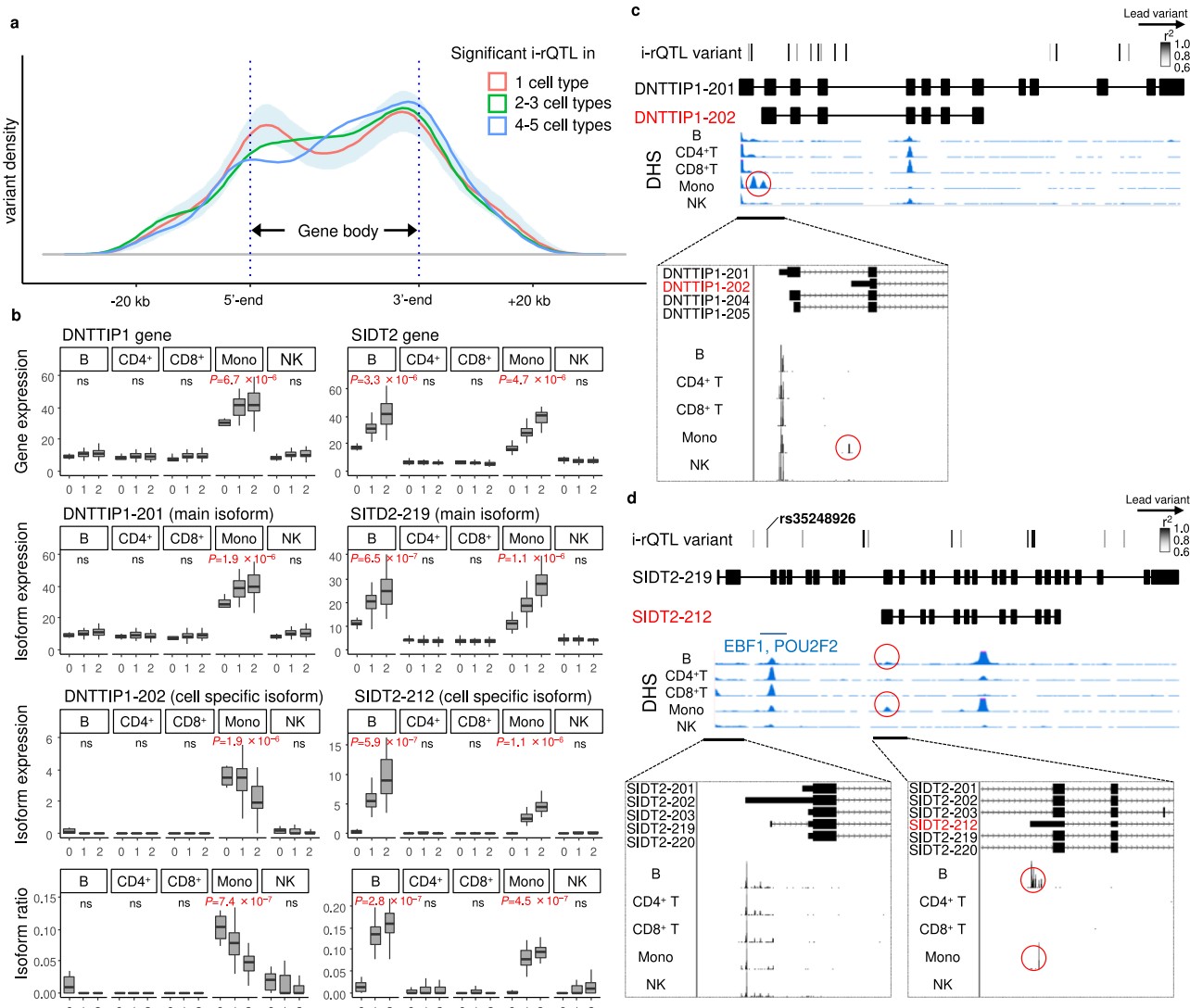

**Fig. 2 | Cell-type specific i-rQTL. a** For each cell-type specificity, distributions of lead i-rQTL variants to the gene body are shown. The shade shows 95% confidence intervals for the mean of the distribution of 1 cell-type specific i-rQTLs. **b** The *DNTTIP1-202* isoform had a monocyte-specific i-rQTL effect. The *SIDT2-212* isoform had B-cell- and monocyte-specific i-rQTL effects. These genes were expressed in all five cell types. On the boxplots the horizontal line indicates the median, the box indicates the first to third quartile of expression and whiskers indicate 1.5 × the interquartile range. Nominal *p*-values obtained by two-tailed tests are shown for

QTL effects which are identified as significant (FDR ≤ 0.05). B; B cells (*n* = 104), CD4⁺; CD4⁺ T-cells (*n* = 103), CD8⁺; CD8⁺ T-cells (*n* = 103), Mono; monocytes (*n* = 105), NK; NK cells (*n* = 104). **c** DHS peaks and FANTOM CAGE peaks of the *DNTTIP1* gene are shown. In monocytes only, there were DHSs and CAGE peaks near the TSS region of the *DNTTIP1-202* isoform. **d** DHS peaks and FANTOM CAGE peaks of the *SIDT2* are shown. In B-cells and monocytes only, there were DHSs and CAGE peaks near the TSS region of the *SIDT2-212* isoform.

levels, and some i-rQTL do not change splicing (e.g., those only altering the transcription start site (TSS)) but consequently change the CDS (e.g., those altering the ATG usage). As previously mentioned, these three types of QTL are not mutually exclusive, but overlapping concepts (Fig. 1b, c).

## Difference in distribution between eQTL and sQTL

We identified 2476–3889 gene eQTL in each subset, which is equivalent to the findings of our prior study[21]. In addition, we identified 3707–6131 i-eQTL as well as 1512–2537 i-rQTL in each cell type (FDR ≤ 0.05, Supplementary Table 1). Next, we evaluated the distribution of the lead variants in each analysis relative to the gene body, which is defined as the region from the TSS to the transcription end site (TES). These three types of QTL were differentially distributed; gene eQTL had a large peak near the TSS. Contrarily, i-rQTL had a larger peak near the TES (A trimodal distribution with two additional peaks in the gene body and TES was detected by Silverman test. *p* = 0.039). i-eQTL

showed an intermediate distribution between the other two QTL (Fig. 1e).

Next, we examined cell-type specificity of i-rQTL effects in commonly expressed genes in five subsets excluding PBL, which contains various cell types. Among a total of 1,703 i-rQTL, 360 i-rQTL showed low cell-type specificity (significant i-rQTL effect observed in four or five subsets) and had a large peak near the TES. In contrast, i-rQTL with high cell-type specificity (significant i-rQTL effect observed in one subset) had a large peak near the TSS (Fig. 2a). As an example of cell-type specific i-rQTL, we showed a monocyte-specific i-rQTL effect on *DNTTIP1-202* and a B-cell- and monocyte-specific i-rQTL effect on *SIDT2-212* (Fig. 2b). These i-rQTL might depend on cell-type specific alternative promoters because DNase hypersensitive sites (DHSs) and CAGE peaks were observed near the TSS in the corresponding subsets. While the lead i-rQTL variant of *DNTTIP1-202* (rs1711194) was located downstream of the gene, several variants in strong linkage disequilibrium (LD, r² ≥ 0.8) were located in the gene body, including near

the CAGE peaks (Fig. 2c). Similarly, because an INDEL variant (rs35248926) in strong LD ($r^2 = 0.91$) with the lead i-rQTL variant of *SIDT2-212* (rs11449159) has eQTL effects on surrounding genes (*TAGLN*, *PCSK7*, *PAFAH1B2* and *SIK3*) in the GTEx dataset[11], this variant may have broad regulatory effects on multiple genes in multiple subsets and may cause the i-rQTL effect. Indeed, B-cell- and monocyte-specific transcription factors such as EBF1 and POU2F2 bind to this region[22,23] (Fig. 2d).

## Integration of shared coding sequence isoforms

To focus on changes in protein sequences caused by sQTL effects, we integrated isoforms with the same CDS by adding their expression levels (FPKM) and then re-analyzed i-eQTL/i-rQTL using these integrated-isoforms (i²-eQTL analysis, integrated-isoform eQTL analysis; i²-rQTL analysis, integrated-isoform ratio QTL analysis). We used the "translation sequences" of the GENCODE v35 basic annotation for isoform integration. A total of 17,050 coding isoforms that share the same CDS with at least one other isoform were integrated into 6724 integrated-isoforms (Fig. 3a). Among them, we identified 2855–4722 i²-eQTL isoforms and 694–1052 i²-rQTL isoforms in each subset (FDR ≤ 0.05, Supplementary Table 2).

To address the methodological advantage of i²-rQTL analysis, we compared the *p*-value of i²-rQTL with the least *p*-value of i-rQTL among the isoforms comprising the corresponding integrated-isoform (Fig. 3b). We identified distinct QTL signals between these two analyses in some genes; in monocytes, while both significant i²-rQTL and i-rQTL effects were observed for 999 genes, only either i²-rQTL or i-rQTL effects (in at least one isoform) were observed for 53 and 302 genes respectively (FDR ≤ 0.05). One of the advantages of i²-rQTL might be its power to detect QTL effects, because integration of isoforms results in a reduction in isoform numbers and an increase in the accuracy of isoform quantification. A typical example was seen in *PARP9* gene; while neither *PARP9-204* or *PARP9-205* with the same CDS had a significant i-rQTL effect, the integrated *PARP9-204 + 205* had a significant i²-rQTL effect (Fig. 3c, d). In contrast, the integrated *ITGB7-201 + 202* did not have a significant i²-rQTL effect, while both *ITGB7-201* and *ITGB7-202* had significant i-rQTL effects (Fig. 3c, d). The splicing events observed in these isoforms altered only the 3′-UTR, and their directions were opposing and mutually exclusive.

Next, we examined the roles of these i²-eQTL and i²-rQTL by evaluating their co-localization with GWAS loci in the GWAS catalog[24] using the RTC Score[25]. In each subset, 674–1197 i²-eQTL and 185–307 i²-rQTL were co-localized with GWAS traits (RTC ≥ 0.8, $r^2 ≥ 0.8$, Supplementary Table 3). For example, i²-rQTL of *PARP9-204 + 205*, as described above, was co-localized with the GWAS of LDL cholesterol levels (rs3762637, RTC = 0.98, $r^2 = 1.00$).

## sQTL analysis for CDS incomplete isoforms with long-read capture RNA-sequencing

To further examine the role of sQTL altering the protein sequences, we focused on CDS incomplete (CDSI) isoforms, which are fragments of CDS registered in the GENCODE comprehensive annotations. These CDSI isoforms were independently defined because they were predicted to have a different CDS compared to the CDS complete isoform (Fig. 4a). Of the 28,075 CDSI isoforms evaluated by our i-rQTL analysis, 175–337 isoforms had i-rQTL effects in each subset (FDR ≤ 0.05). Among them, 59–129 CDSI i-rQTL were co-localized with GWAS in each subset (RTC ≥ 0.8, $r^2 ≥ 0.8$, Supplementary Table 4), suggesting that these CDSI would have biological functions and be involved in disease pathogenesis.

Next, we determined the sequences of the full-length isoforms using long-read capture RNA-seq. As mentioned above, i-rQTL analysis can detect differences in TSS compared to junction-based sQTL analysis but may produce false positives due to inaccuracies in isoform quantification. However, 43–46% of i-rQTL signals were overlapped

with those of LeafCutter analysis[9,20] (FDR ≤ 0.05, Supplementary Fig. 1). We selected 37 CDSI isoforms of which i-rQTL signals were confirmed by LeafCutter analysis on the junctions specific to CDSI isoforms. We used xGen custom probes to capture these isoforms. We identified a candidate list of full-length isoforms using the FLAIR pipeline[26]. From this list, we extracted isoforms whose 5′- and 3′-end were within 50 bp of known TSSs and TESs[27,28] and whose high coding probability was confirmed by CPAT[29] (Fig. 5a). In a comparison between long-read capture RNA-seq and conventional long-read RNA-seq using LCL (B-cells) or THP-1 (monocytes), the former detected 1.8-fold more full-length of "CDS complete" isoforms on average, even though the total number of reads was less than one-tenth (Fig. 5b). As a result of analyses using six cell lines of LCL, THP-1, Jurkat (T-cells), HEK293 (embryonic kidney cells), HepG2 (liver cells), and K562 (leukaemic cells), we identified full-length of CDSI isoforms in 78.4% of targeted genes (29 out of 37 genes). Of note, we identified multiple full-length isoforms for corresponding CDSI isoforms in most genes (90.0%, 26/29). The variation of completed CDSI isoforms were different among these genes: the number of isoforms accounting for over half of all completed CDSI isoforms was one for 22 genes, two for three genes, three for one gene, and six for two genes. The remaining gene, *ATXN2L*, which had many known splicing isoforms, required 26 isoforms (Supplementary Fig. 2).

Since we confirmed the presence of completed CDSI isoforms at the transcriptional level but not at the protein level, we evaluated the translation of these isoforms using PeptideAtlas[30], a mass spectrometry-based peptide database. Of the 37 CDSI isoforms, we identified 4 isoforms having peptides unique to the isoforms, indicating they were indeed translated (Supplementary Table 5). Although peptides translated by CDSI isoforms would be difficult to detect by mass spectrometry due to their relatively low expression levels, we confirmed the translation of 1461 isoforms out of 26,457 CDSI isoforms registered in GENCODE v35.

Among the 29 genes confirmed for their full-length isoforms, the *BST1* gene is a GWAS candidate gene for Parkinson's disease (PD). The i-rQTL effect on *BST1-205* results in an alternative final coding exon relative to the main isoform, *BST1-201*. The lead i-rQTL variant (rs4263397) was located only 18 bp from this alternative splice acceptor site (SAS) (Fig. 4c). The GWAS risk allele increased the ratio of this isoform, while decreasing the overall expression of *BST1* gene (Fig. 4d). BST1 is a glycosyl-phosphatidylinositol (GPI)-anchored membrane protein with a short hydrophobic region at the C-terminus, which forms a dimer[31,32]. According to the predicted protein structure by AlphaFold2[33,34], the homodimer of BST1-201 and the heterodimer (BST1-201 and BST1-205) had the same structure except for the C-terminus (Fig. 5e, RMSD = 0.067).

The *CARD9* gene is a GWAS candidate gene for ankylosing spondylitis (AS) and inflammatory bowel disease (IBD). The i-rQTL effect on *CARD9-208* shortened the 5′-side of exon 2 by 124 bp. The lead i-rQTL variant (rs4498662) was located downstream of the gene, but the variant in strong LD (rs10781499, $r^2 = 0.90$) was located only 19 bp from this alternative SAS. This isoform had a downstream alternative first ATG, or translation start site, and consequently lacked the CARD domain (Fig. 4c). The risk allele of AS decreased the ratio of *CARD9-208* and increased that of *CARD9-201* (Fig. 4d). CARD9 has a CARD domain at its N-terminus that mediates interactions between CARD-containing molecules, and a coiled-coil at its C-terminus that functions as an oligomerization domain[35]. According to the predicted structures, both CARD9-201 and CARD9-208 had a coiled-coil domain and their structures were similar (RMSD = 4.7 Å), and they could oligomerize similarly through this domain (RMSD = 7.1 Å, Fig. 5d, f).

The *ATXN2L* gene is a GWAS candidate gene for intelligence. Ataxin type 2 associated protein encoded by the *ATXN2L* gene is a member of the spinocerebellar ataxia (SCA) family associated with neurodegenerative diseases. The i-rQTL effect was observed on the

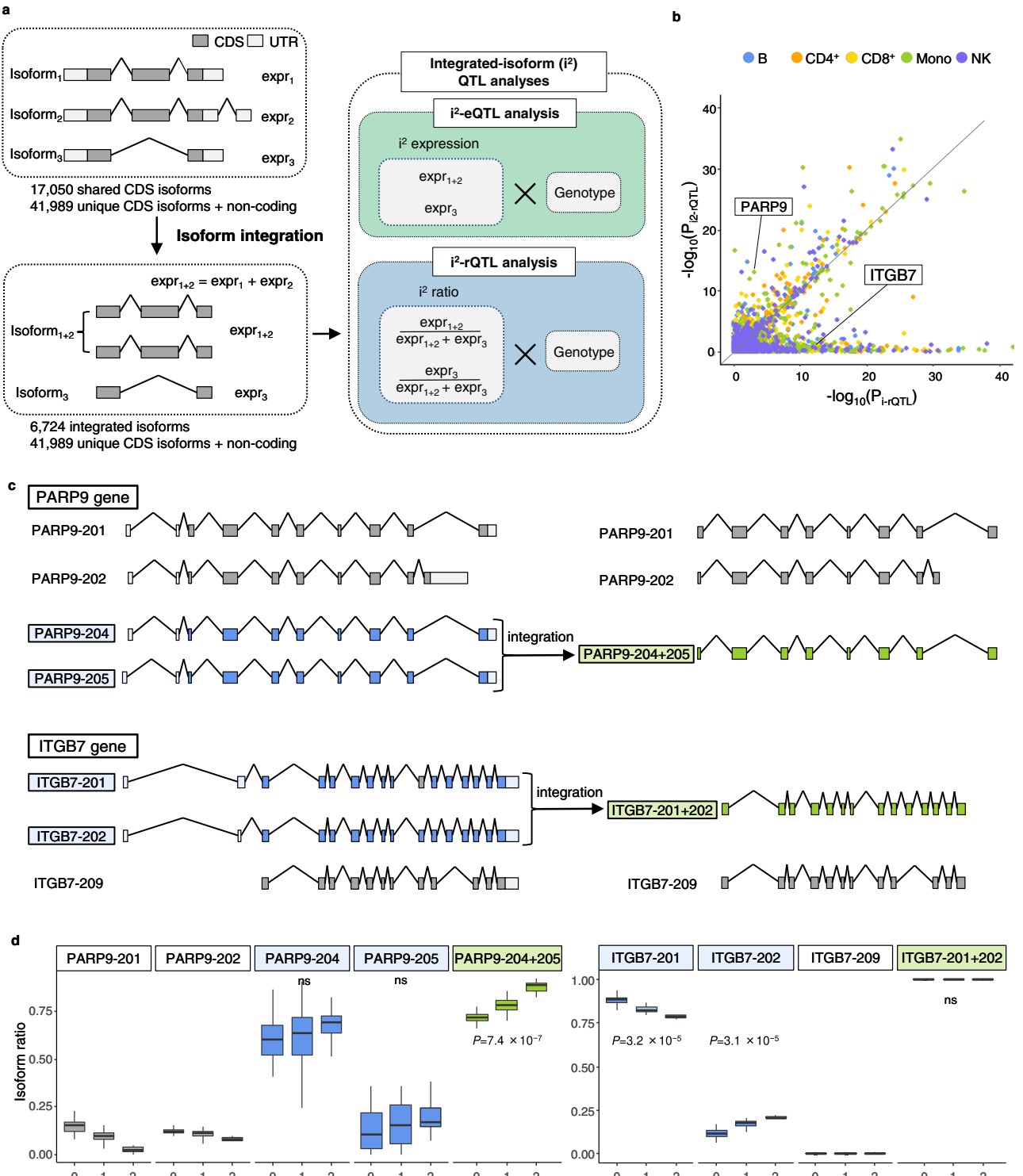

**Fig. 3 | Integration of isoforms (i²) by shared CDS; i²-eQTL analysis and i²-rQTL analysis. a** There are three isoforms of gene A: isoforms 1, 2, and 3. Isoforms 1 and 2 have the same CDS. Expressions of isoforms with the same CDS were combined and used for correlation analysis (i²-eQTL analysis, integrated-isoform eQTL analysis; i²-rQTL analysis, integrated-isoform ratio QTL analysis). **b** Comparison of the *p*-values of the correlation analysis before and after the integration of isoforms. The *p*-values of isoforms before integration were the minimum of the i-rQTL *p*-values of each integrated isoform. **c, d** *PARP9-204* and *PARP9-205* have the same CDS and were not

identified as significant i-rQTL in monocytes (*n* = 105). The integrated isoform (*PARP9-204 + 205*) was identified as a significant i²-rQTL. In contrast, *ITGB7-201* and *ITGB7-202* had the same CDS and were identified as significant i-rQTLs in NK cells (*n* = 104). The integrated isoform (*ITGB7-201 + 202*) was not identified as a significant i²-rQTL. On the boxplots the horizontal line indicates the median, the box indicates the first to third quartile of expression and whiskers indicate 1.5 × the interquartile range. Nominal *p*-values obtained by two-tailed tests are shown for QTL effects which are identified as significant (FDR ≤ 0.05).

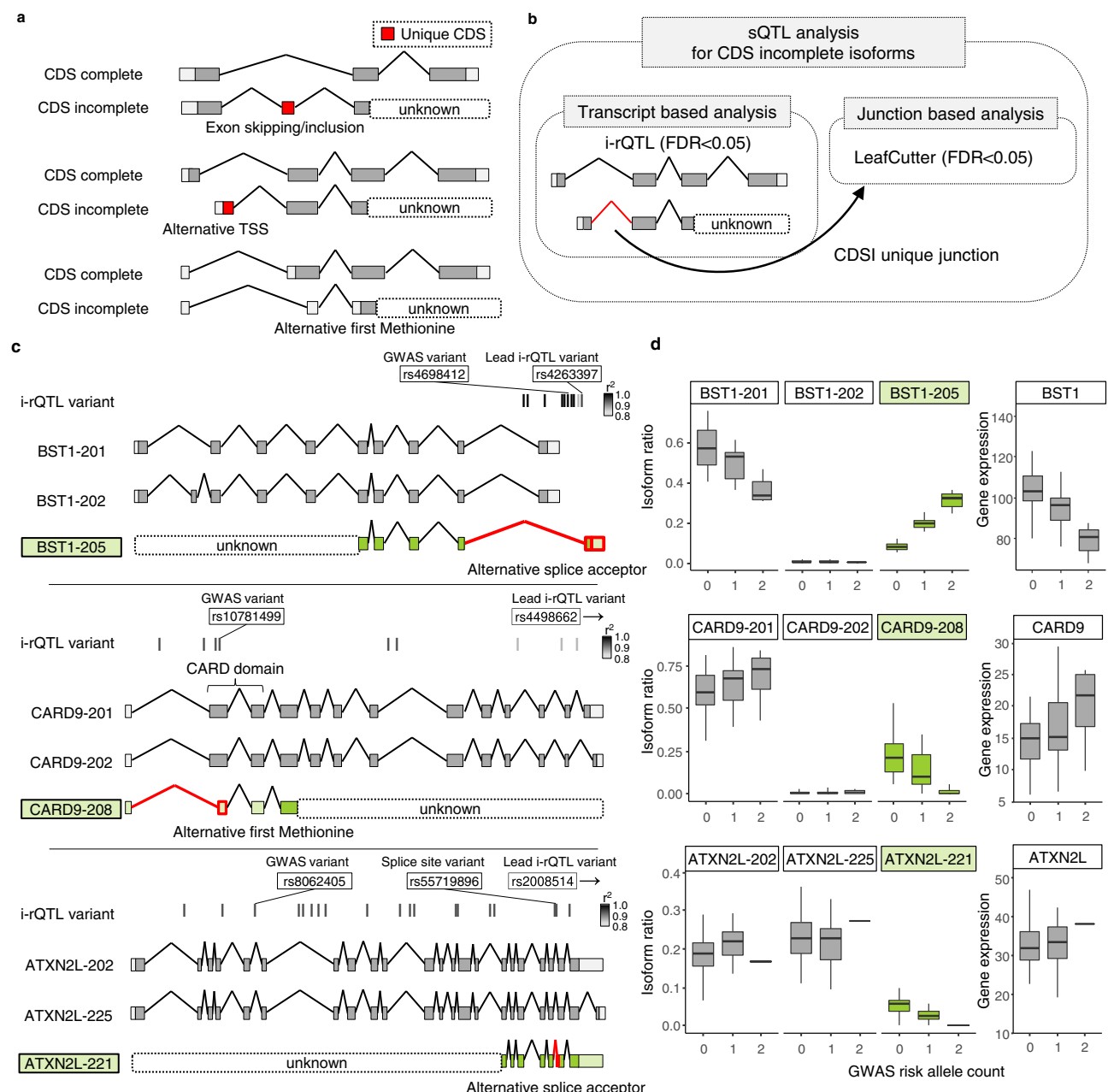

**Fig. 4 | sQTL analysis for CDS incomplete isoforms. a** Examples of CDS incomplete isoforms with and without unique CDS. **b** sQTL analysis for CDS incomplete isoforms. **c** Gene models of the major isoforms of *BST1*, *CARD9* and *ATXN2L*. *BST1-205*, *CARD9-208* and *ATXN-221* have incomplete CDS. **d** Boxplots of i-rQTL effects on CDS incomplete isoforms of three genes and eQTL effects on those genes. On the boxplots the horizontal line indicates the median, the box indicates the first to third quartile of expression and whiskers indicate 1.5 × the interquartile range.

*ATXN2L-221* isoform, of which the second exon from the 3'-end was extended by 3 bases. The lead i-rQTL variant (rs2008514) was located downstream of the gene, but the variant in strong LD (rs55719896, $r^2 = 0.96$) was located on the alternative SAS (Fig. 4c). Long-read capture RNA-seq identified over 60 isoforms with this splice junction. Notably, the majority of mRNAs with this *ATXN2L-221*-specific SAS adopted an alternative TSS (Fig. 5g, $p = 1.91 \times 10^{-39}$, chi-squared test) and contained another splice junction within the final exon (> 90%, $p = 6.00 \times 10^{-50}$, chi-squared test). This may reflect the thermodynamic stability of the secondary structure predicted by the RNAfold of the four isoforms from two different TSSs with two different SASs. (Supplementary Figure 3). According to the predicted protein structures, ATXN2L-221 and ATXN2L-203 had dissimilar structures (RMSD = 27.0 Å, Fig. 5h).

## Evaluation of trans-eQTL effects via sQTL effects with protein structural changes

Finally, we assessed the biological importance of CDS changes introduced by sQTL, utilizing trans-eQTL effects (Fig. 6a). While $i^2$-rQTL by definition cause changes in CDS, some i-rQTL cause changes in CDS and some do not. Therefore, we compared the trans-eQTL effects of $i^2$-rQTL and i-rQTL that do not change CDS. We here defined that the trans-eQTL effect was the effect of a sQTL variant on the expression levels of distal genes. For instance, if a sQTL variant affected the ratio of two isoforms that were translated into two protein isoforms of a transcription factor, then the sQTL variant would affect the expression of distal genes that would be differentially regulated by these two transcription factor isoforms. Comparing 5 million randomly selected variant-gene pairs, the QQ-plot analysis indicated enrichment of the

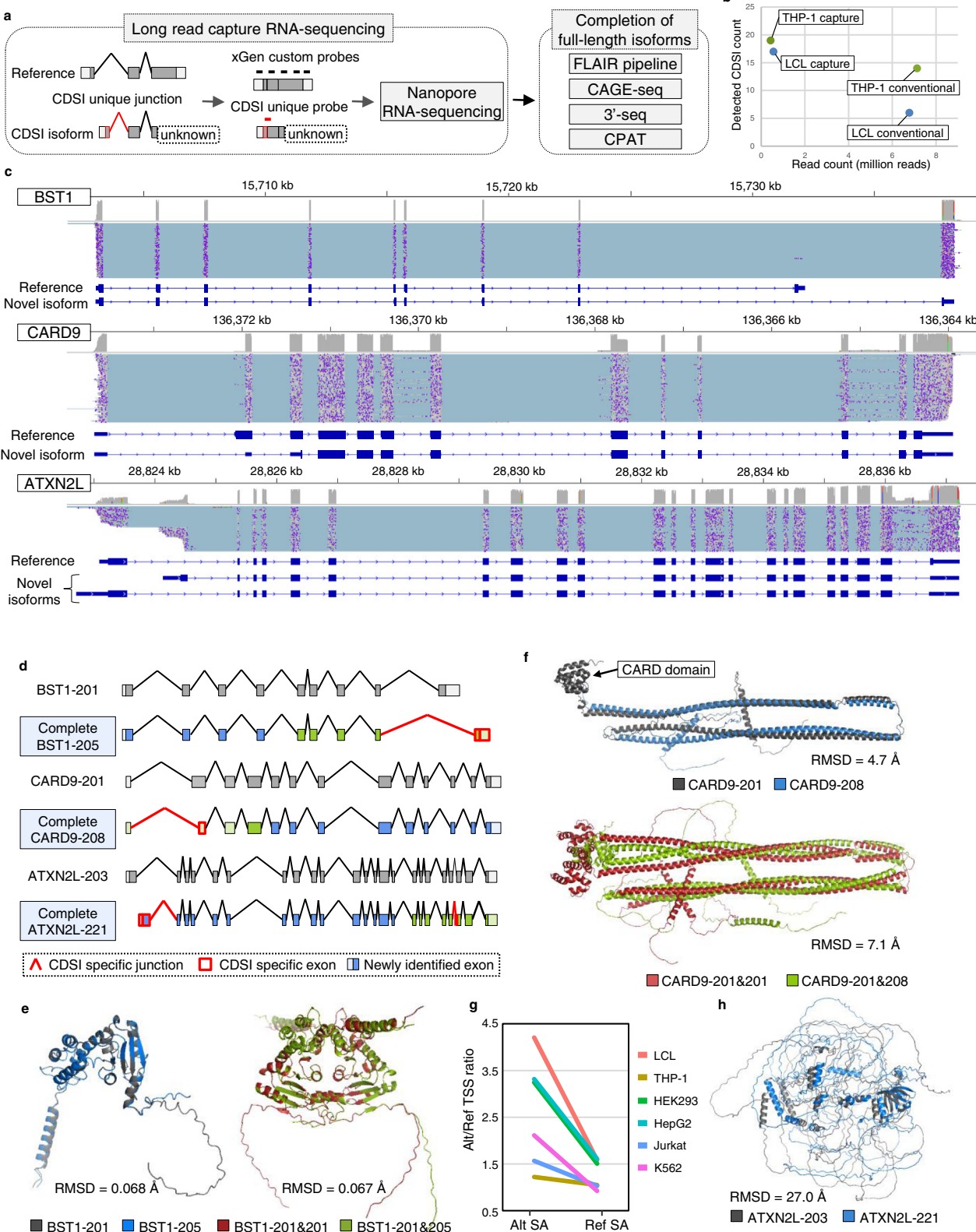

significant trans-eQTL effect in i²-rQTL variants (Fig. 6b), which warranted the following analysis.

The trans-eQTL effects of sQTL variants may give clues to the mechanisms of disease, and we here took an example of the GWAS loci for systemic lupus erythematosus (SLE) at 6p21.31. This locus had a large LD block[36] including multiple eQTL and i²-rQTL genes (Fig. 6c). Previous studies[37,38] had considered the *UHRF1BP1* gene as candidate

causal gene because of its strong eQTL effect. The present analysis indicated the *SNRPC* gene as another candidate gene, because it had a strong i²-rQTL effect with protein structural changes. While the main isoform (*SNRPC-201*) of the *SNRPC* gene has a zinc-finger domain, the minor isoform (*SNRPC-203*) does not (Fig. 6d). These *SNRPC* isoforms had an i²-rQTL effect in all subsets (Fig. 6e). The GWAS risk allele increased the isoform ratio of *SNRPC-201* and decreased that of *SNRPC-*

**Fig. 5 | Long-read capture RNA-sequencing for CDS incomplete isoforms.**
**a** Overview of long-read capture RNA-sequencing analysis for CDS incomplete
isoforms. **b** Comparison of long-read capture RNA-sequencing and conventional
long-read RNA-sequence. **c** Results of long-read capture RNA-sequencing for three
genes. A compressed light blue horizontal line in the alignment track corresponds
to a single read. Gene models are shown for the reference isoform and major
unannotated isoform(s) with CDS incomplete isoform specific junction.
**d** Comparison of reference isoforms and completed CDS incomplete isoforms.

**e** Predicted protein structures using ColabFold; BST1-201 and BST1-205, BST1
homodimer (BST1-201) and heterodimer (BST1-201 and BST1-205). **f** Predicted
protein structures using ColabFold; CARD9-201 and CARD9-208, CARD9 homo-
dimer (CARD9-201) and heterodimer (CARD9-201 and CARD9-208). **g** The ratio of
*ATXN2L* isoforms with alternative TSS to those with reference TSS for each isoform
with reference or alternative splice acceptor. **h** Predicted protein structures using
ColabFold; ATXN2L-225 and ATXN2L-221. CDSI CDS incomplete, TSS Transcription
start site, SA Splice acceptor.

*203*. We evaluated the co-localization of GWAS and these QTL signals,
and the RTC score of $i^2$-rQTL variant (rs1180775074) on the *SNRPC* gene
was 0.90–0.98 in each subset, while that of eQTL variant (rs2764197)
on the *UHRF1BP1* gene was 0.7–0.75 (Fig. 6f). Because the trans-eQTL
effect of this $i^2$-rQTL variant on each distal gene was modest, we
evaluated the effects collectively by using Gene Set Enrichment Ana-
lysis (GSEA)[39,40]. An increase in the isoform ratio of *SNRPC-201* and a
decrease in that of *SNRPC-203* enhanced the expression of interferon
signature genes (ISGs), which are hallmarks of SLE patients[41]. Inter-
estingly, this tendency was remarkable in men in the stratified analysis
by gender of the samples (Fig. 6g).

## Discussion

The landscape of mRNA isoforms is much more complex than expec-
ted. Indeed, an analysis of 9795 RNA-seq samples from the GTEx col-
lection identified 266,331 transcripts in protein-coding genes, which
doubled the number of transcripts registered in the RefSeq database[42].
Therefore, we needed to establish an efficient method to find disease-
causing isoforms among the myriad of isoforms with or without phy-
siological functions. In the present study, we proposed two strategies,
integration and selection, for this purpose. For the former, we per-
formed $i^2$-rQTL analysis to integrate isoforms with the same CDS, and
for the latter, we selected CDS incomplete isoforms and confirmed
them by long-read capture RNA-seq.

We adopted $i^2$-rQTL analysis as the main approach to identify
sQTL, instead of junction-based sQTL analysis. This analysis can iden-
tify the full-length of sQTL isoforms, which is important for assessing
functional changes in proteins. In addition, this analysis can also detect
sQTL effects in a broad sense including those caused by alternative TSS
or alternative TES (alternative PAS usage) without changes in junc-
tions. Although the previous studies showed less cell-specific effects of
sQTL compared to eQTL[2], our observation that sQTL effects caused by
alternative TSS tended to have cell-type specificity (Fig. 2) suggested
that previous junction-based sQTL studies might have underestimated
the cell-type specific sQTLs and their roles in diseases. However, it is
true that the junction-based sQTL analysis has improved the robust-
ness of sQTL analysis[20], because it not only skips the process of
transcript-assembly but also integrates isoforms with the same junc-
tion. We apply the latter advantage to $i^2$-rQTL analysis by integrating
isoforms with the same CDS ($i^2$-rQTL analysis) to identify biologically
important sQTL, which was supported by our finding that $i^2$-rQTL
affected the expression of more genes than i-rQTL without $i^2$-rQTL
effects (Fig. 6b).

For the second strategy, we performed i-rQTL analysis focusing on
CDSI isoforms. We further confirmed the full-length sequences of
these incomplete isoforms by long-read capture RNA-seq, which
appeared to be more efficient compared to conventional long-read
RNA-seq (Fig. 5b). Through this analysis, we demonstrated that low
expression isoforms like CDSI isoforms might indeed be associated
with diseases. For the first example, the PD risk allele rs4698412-A
increased *BST1-205*, which has an alternative coding exon 9 corre-
sponding the GPI-anchor domain, compared to *BST1-201* (Fig. 5d). The
selective vulnerability of dopaminergic neurons in PD is caused by a
$Ca^{2+}$ imbalance[43]. BST1 exists as a dimer protein and releases $Ca^{2+}$ from
intracellular $Ca^{2+}$ stores via the production of cyclic ADP-ribose[32]. The
altered GPI-anchor domain of BST1-205 may influence the protein

function, possibly with an inhibitory or dominant-inhibitory effect on
BST1-201 (Fig. 5e). For the second example, the AS risk allele
rs10781499-A decreased CARD9-208 lacking the CARD domain, which
mediates interactions between CARD-containing molecules. As
CARD9-208 retains an oligomerization domain, it may have an inhi-
bitory (or dominant inhibitory) effect on its main isoform as observed
with BST1-205. Because opposing QTL effects were observed with the
main isoforms (*BST1-201* and *CARD9-201*), the total increase (*BST1*) or
decrease (*CARD9*) in gene function in the risk allele could be syner-
gistic if the spliced isoforms have dominant inhibitory effects. For the
third example, the effective allele of IBD, BMI, asthma, and type 2
diabetes and intelligence, rs8062405-G decreased the expression of
*ATXN2L-211*. Though this locus showed multiple eQTL/i-rQTL effects
for multiple genes, the co-localization of the i-rQTL effect on *ATXN2L-
221* with the GWAS signal is more robust in all subsets, suggesting that
*ATXN2L-221* is responsible in this locus. The stronger sQTL effect on
alternative SAS may be the primary event at this locus, but how it alters
the TSS usage remains unclear. However, the amount of isoforms with
alternative TSS and alternative SAS may reflect differences in mRNA
stability. Therefore, not only the amino-acid change in the junction but
also the secondary change in the N-terminal domain of *ATXN2L-221*
may be responsible for the disease.

While most GWAS loci have a single functional variant such as a
cis-eQTL variant, some GWAS loci have multiple functional variants
(missense, eQTL/sQTL variants) and multiple candidate genes, which
make it difficult to determine the disease-causing gene. Similar to the
roles of driver and passenger mutations in cancer cells, some of the
variant effects are true disease-causing, while others are not disease-
related; however, distinguishing between them is difficult. We esti-
mated the candidate causal QTL effect by examining trans-eQTL effect,
based on the assumption that $i^2$-rQTL effects on CDS would sub-
stantially influence gene function. This approach successfully identi-
fied a candidate causal gene in the SLE risk loci, in which the disease
risk allele (rs2764208-G) had an $i^2$-rQTL effect on *SNRPC* isoforms.
*SNRPC* encodes the U1-C protein, which is one of the components of
U1-snRNP (small nuclear ribonucleoprotein) and is known to be an SLE
specific autoantigen. The U1-snRNP immune complex induces inter-
feron signaling via TLR7 in plasmacytoid dendritic cells, whose sig-
naling is different in the presence and absence of estrogen[44].
Considering the correlation between the isoform ratio and the
induction of ISGs expression, SNRPC is a strong candidate for the
causal genes in this locus. In addition, the sex-biased results observed
in GSEA may lead to the elucidation of gender differences in the
pathophysiology of SLE.

Our analyses have several limitations. Firstly, we might have
underestimated the effects of alternative UTR usages as a result of our
focus on the CDSs. Accumulating evidence has shown that both 5′- and
3′-UTRs have effects on the stability of mRNA or its translational effi-
cacy, through the binding of miRNA or RNA-binding proteins[45,46].
Indeed, with regards to disease genetics, a recent report has shown
that genetic variants affecting alternative polyadenylation signals
(PASs) explained a substantial proportion of heritability for complex
diseases, emphasizing the importance of 3′-UTR[47]. Secondly, we used
sQTL data obtained from a Japanese population in the co-localization
analysis of GWAS loci, which were mostly derived from a European
population. These population differences could bias our findings,

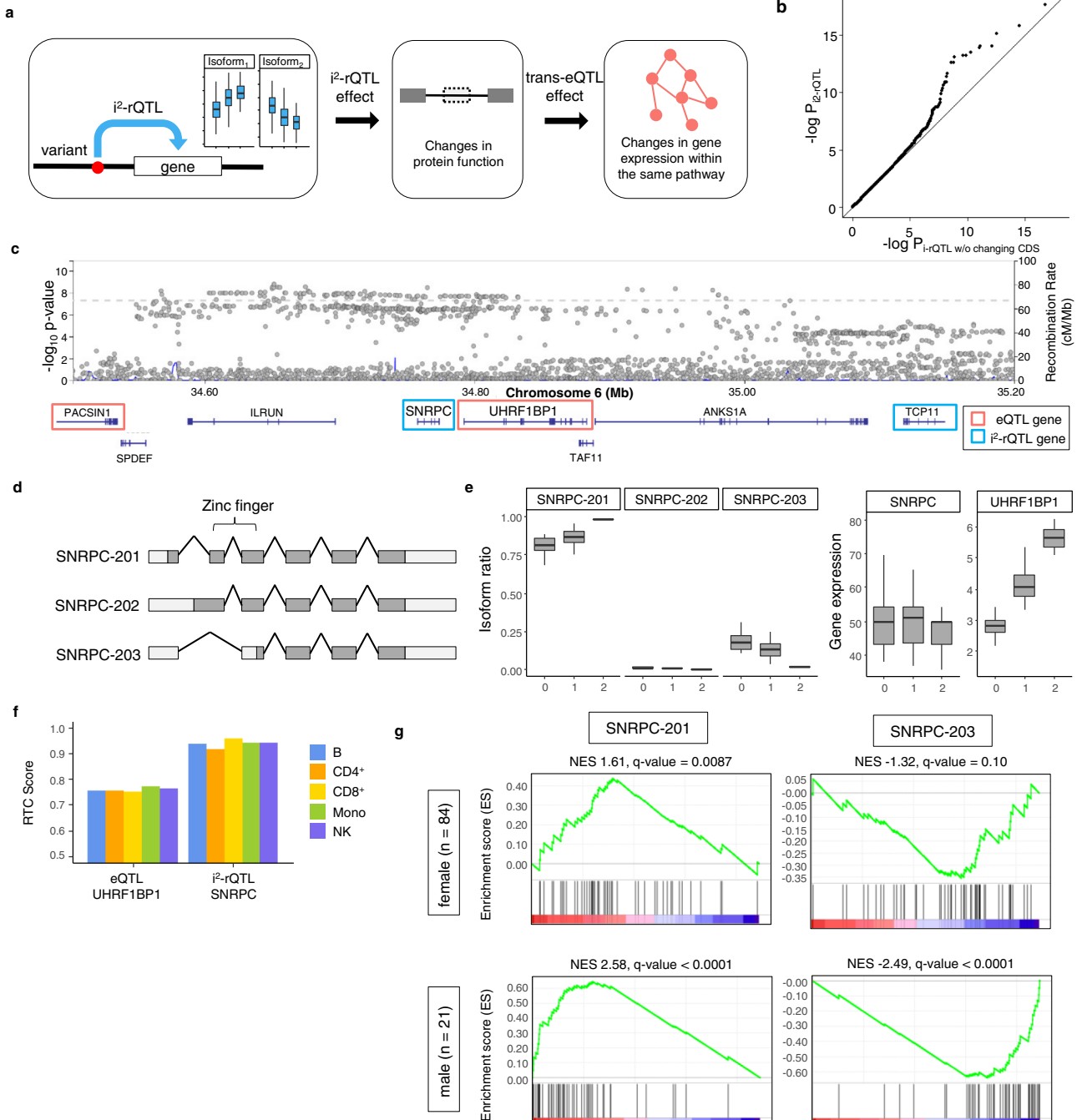

**Fig. 6 | Identification of causal genes using trans-eQTL analysis at genomic loci with multiple eQTL and i²-rQTL effects. a** Schematic diagram of how an i²-rQTL effect leads to a trans-eQTL effect. **b** Comparison of trans-eQTL *p*-values of i-rQTL variants and those of i²-rQTL variants. **c** The *SNRPC-UHRF1BP1* locus forms a large linkage disequilibrium block and contains multiple eQTL or i²-rQTL genes. **d** Gene models of major protein coding isoforms of the *SNRPC* gene. **e** i²-rQTL effects on *SNRPC* and eQTL effect on *UHRF1BP1* of the SLE GWAS lead variant (rs2764208) in monocytes (*n* = 105). On the boxplots the horizontal line indicates the median, the box indicates the first to third quartile of expression and whiskers indicate 1.5 × the interquartile range. **f** RTC scores for SLE GWAS of i²-rQTL effect on *SNRPC* and eQTL effect on *UHRF1BP1*. **g** Gene Set Enrichment Analysis (GSEA) using Interferon Signature Genes (ISGs) for *SNRPC* isoform ratios in monocytes in both sexes.

especially in the direction of false-negative co-localization. However, more than 80% of i²-rQTL effects in B-cells from the Japanese population in our analysis were validated in LCLs (*n* = 373) from the European population[7] (FDR ≤ 0.05), and 96.4% of these effects were consistent in their direction (Supplementary Fig. 5). This indicated that most sQTL were shared between these populations. Thirdly, our analysis could not deny the possibility of disease-causing effects of other functional variants, such as missense variants or eQTL variants, existed on the same haplotype of i-rQTL/i²-rQTL. For example, in the *ATXN2L* locus

having multiple eQTL/i-rQTL effects for multiple genes, the eQTL effect on *IL27* could be responsible for IBD, because the anti-inflammatory roles of IL-27 have been established in a murine colitis model[48]. Furthermore, because multiple functional variants having effects on the same gene may simultaneously cause the same disease, which occasionally occurs in complex traits[49], it may be difficult to exclusively evaluate the roles of sQTL variants in diseases.

With the advent of long-read sequencing technologies, alternative splicing in disease etiology has gained more attention in recent years.

Although several studies have established isoform catalogues in various tissues and cells using long-read RNA-seq[50], there remains many isoforms unidentified due to lack of read depths. Instead, we proposed long-read capture RNA-seq for these low-expressed and poorly-annotated isoforms, combined with sQTL analysis focusing on CDS. This approach would be powerful for identifying true causal effects in the GWAS loci, and functional dissection of identified isoforms in animal models should elucidate unknown mechanisms of disease pathogenesis.

## Methods

### eQTL dataset

We reanalyzed short-read RNA-seq data of five immune subsets and peripheral blood leukocytes (PBL) from 110 Japanese healthy volunteers[21]. Of the volunteers, 88 (80%) were females and the average age was $39.0 \pm 10.6$ years. TruSeq Stranded mRNA Library Prep Kits (Illumina) were used for the RNA-Seq library preparation, and a HiSeq 2500 was used for sequencing (paired-end 125 bp reads).

Infinium OmniExpressExome BeadChip (Illumina) was used for genotyping. Genotype imputation was done using Minimac3[51] and 1000 Genomes Project Phase 3 (release 3) as a reference panel. We used 5,415,012 SNPs and 718,459 InDels on autosomes with MAF ≥ 0.05 for analyses. Detailed methods of the eQTL dataset preparation are available in our previous paper[21].

### Read mapping and transcriptome assembling

We re-conducted the read mapping and transcriptome assembling from our previous study[21] using different tools and updated references. The mapping tool was changed from TopHat2 to STAR v2.7.5a. We also used the updated reference genome (GRCh38) and transcript annotation (GENCODE v35 comprehensive and basic annotation). Transcriptome assembling and quantification of expression levels were performed by Cufflinks v2.2.1, using reads of uniquely mapped and concordant alignment.

### eQTL analysis

We performed independent eQTL analysis for each cell type using QTLtools v1.3.1[52]. We defined the analysis using the expression level of genes as "gene eQTL analysis", and the analysis using the expression level of each isoform as "isoform eQTL analysis". We performed the following two types of isoform eQTL analysis: 1. analysis using GEN-CODE v35 comprehensive annotation (i-eQTL; isoform eQTL analysis), and 2. analysis focusing on changes in protein structure using GEN-CODE v35 basic annotation as below. In the second analysis, we grouped isoforms with the same CDS and summed the expression levels of these isoforms. We used the grouped isoform as a single isoform for the following analysis ($i^2$-eQTL; integrated-isoform eQTL analysis). In gene eQTL analysis, genes with gene FPKM > 0.1 in all samples of each cell type were analyzed. In i-eQTL/$i^2$-eQTL analysis, isoforms of these gene with isoform FPKM > 1 in 5% samples were analyzed. We performed quantile normalization, then rank-transformed normalization, and finally PEER normalization[53] using 15 hidden factors to eliminate the batch effect between experiments. Correlation analysis with the genotype was performed on the SNP with MAF ≥ 0.05 existing within the gene-SNP distance ≤ 1 MB using permutation pass of QTLtools. In order to correct multiple hypothesis testing, we used the Storey & Tibshirani False Discovery Rate procedure implemented in the R/qvalue package.

### sQTL analysis

We identified sQTLs by i-rQTL (isoform ratio QTL, trQTL) analysis[54]. i-rQTL analysis is an isoform based sQTL analysis. We performed the following two types of analysis similar to i-eQTL/$i^2$-eQTL analysis: 1. analysis using GENCODE v35 comprehensive annotation (i-rQTL analysis), and 2. analysis focusing on changes in protein structure using GENCODE v35 basic annotation ($i^2$-rQTL; integrated-isoform ratio QTL analysis). We calculated the expression ratio of the isoform over all isoforms and normalized in the same way as for eQTL analysis. We also conducted junction-based sQTL analysis using LeafCutter[20]. LeafCutter has high sensitivity for detecting sQTL junctions. We used a series of scripts available at the LeafCutter website. We counted the uniquely mapped junction reads using bam2junc.sh. We then clustered the junction read counts using leafcutter_cluster.py. The maximum intron length was set to 500,000 bp, and the minimum number of reads of the cluster was set to 50 reads. We normalized the junction read ratio in the same cluster in the same way as for eQTL analysis.

### Co-localization analysis of GWAS and QTL signals

We evaluated the co-localization of GWAS and QTL effects using RTC (regulatory trait concordance) score[25]. Calculation of RTC score was performed using QTLtools RTC mode. For GWAS data, we used the EMBL-EBI GWAS catalog[24] as of August 26, 2020. In integrating srQTL analysis and GWAS, we used LeafCutter[20] for curation of i-rQTL/i2-rQTL analysis results by the following criteria: 1. FDR; false discovery rate ≤ 0.05 in the i-rQTL/i2-rQTL analysis, 2. FDR ≤ 0.05 in the Leaf-Cutter analysis, and 3. the direction of sQTL effects (calculated as beta) were consistent in both analyses.

### Long-read capture RNA-seq for CDSI isoforms

We conducted long-read RNA-sequencing for the CDSI isoforms (37 isoforms in total), whose i-rQTL signals were co-localized with disease GWAS signals and whose unique splice junctions showed significant sQTL signals in LeafCutter analysis (FDR ≤ 0.05). We prepared xGen Custom Target Capture Probes (biotinylated 120bp-ssDNAs generated by IDT) that covered the entire main-isoform sequences of corresponding genes as well as the unique junction sequences for the CDSI isoforms. All 1,411 sequences of custom target probes are available in the Supplementary Data 7. We isolated total RNA from six cell lines, LCL, THP-1, Jurkat, HEK293, HepG2, and K562 (ATCC; American Type Culture Collection) using TRIZOL Reagent (ThermoFisher), RNeasy Mini Kit (QIAGEN) and RNase-Free DNase Set (QIAGEN). We reverse transcribed 100 ng of total RNA by smartseq v2 protocols[55] with oligo-dT primers and then amplified them by 22 cycles of PCR using KAPA HiFi Hot Start Ready Mix (Kapa Biosystems) with 5Me-isodC-TSO and ISPCR primers. We hybridized and captured the cDNA with xGen probes using an xGEN Hybridization and Wash Kit (IDT) according to the manufacturer's protocol. We then amplified the captured cDNA with additional 10 cycles of PCR as described above. For library pre-paration for sequencing, we used a Nanopore Ligation Sequencing Kit (SQK-LSK109; Oxford Nanopore Technologies) and NEBNext Quick Ligation Module/NEBNext Ultra II End-Repair/dA-Tailing Module (New England Biolabs). Then cDNAs were sequenced by MinION (Oxford Nanopore Technologies) with a Flongle Flow Cell (FLO-FLG001). Basecalling was done using Guppy (v4.4.1). The obtained fastq files were aligned to the GRCh38 primary assembly using minimap2 with reference to the splice junctions in the GENCODE v35 annotation. After flair-correct and flair-collapse, we extracted isoforms whose 5'-end was located within 50 bp from the FANTOM CAGE peaks (relaxed TSS) and whose 3'-end was located within 50 bp from the TES of PolyASite2.0. The ORFs of isoforms were identified using CPAT (coding probability ≥ 0.364).

We performed conventional long-read RNA-seq using 300 ng of total RNA from LCL and THP-1. We used the Magnosphere UltraPure mRNA Purification Kit (TAKARA BIO) to isolate mRNA. After reverse transcription and switching reactions using a cDNA-PCR sequencing kit (SQK-PCS109; Oxford Nanopore Technologies), we amplified the cDNA by 10 cycles of PCR with LongAmp Taq (New England Biolabs). We then sequenced them using GridION X5 (Oxford Nanopore Technologies) with a MinION Flow Cell (R9.4.1/FLO-MIN106D). Data processing was done as for the capture RNA-seq.

## Protein structure prediction

We extracted the longest ORF sequence from the full-length mRNA sequence identified by long-read capture RNA-seq using ORFfinder (NCBI). After converting the nucleotide sequence to amino acid sequence according to the codon table, we confirmed that the reading frame matched the corresponding isoform translation sequence in gencode v35 comprehensive annotation. We used ColabFold2[34], which is based on AlphaFold2[33], a tool for predicting higher-order structures of proteins using deep learning, to predict the structure of the protein or complex with the amino acid sequence. All options were used as default (use_amber = False, use_templates = False, msa_mode = MMseqs2 (UniRef+Environmental), model_type = auto (AlphaFold2-ptm and AlphaFold2-multimer), pair_mode = unpaired + paired, num_recycles = 3).

## Trans-eQTL analysis

To compare the strength of the effect of i-rQTL and $i^2$-rQTL to alter the expression of other genes (trans-eQTL effect), we performed correlation analysis between lead i-rQTL and $i^2$-rQTL variants and expression levels of other genes (excluding the genes having cis-eQTL effects, FDR ≤ 0.05). In addition, we applied Gene Set Enrichment Analysis (GSEA[39,40]) to evaluate the effect of $i^2$-rQTL variants on the expression of specific gene sets. The normalized isoform ratio of the $i^2$-rQTL gene, which is a continuous variable, was used as the phenotype label, and the normalized FPKM of the gene was used as the expression data. Pearson's correlation coefficient was used for weighted metrics.

## URLs

Minimac3 (https://genome.sph.umich.edu/wiki/Minimac3)
PEER (http://www.sanger.ac.uk/science/tools/peer/)
STAR (https://github.com/alexdobin/STAR/)
Cufflinks (http://cole-trapnell-lab.github.io/cufflinks/)
QTLtools (https://qtltools.github.io/qtltools/)
LeafCutter (https://github.com/davidaknowles/leafcutter)
1000 Genomes Project (http://www.1000genomes.org/)
GENCODE Project (http://www.gencodegenes.org/)
GWAS catalog (https://www.ebi.ac.uk/gwas/home/)
FLAIR (https://github.com/BrooksLabUCSC/flair/)
CPAT (https://cpat.readthedocs.io/en/latest/)
ColabFold (https://github.com/sokrypton/ColabFold)
ORFfinder (https://www.ncbi.nlm.nih.gov/orffinder/)
RNAFold (http://rna.tbi.univie.ac.at/cgi-bin/RNAWebSuite/RNAfold.cgi)
GSEA (https://www.gsea-msigdb.org/gsea/)

## Reporting summary

Further information on research design is available in the Nature Research Reporting Summary linked to this article.

## Data availability

The main results of $i^2$-rQTL analysis are available in Supplementary Data 1, 2. Summary statistics are available at Synapse (Accession no. syn33245388; https://doi.org/10.7303/syn33245388). The main results of i-rQTL analysis for CDS incomplete isoforms are available in Supplementary Data 3, 4. The completed gene models for CDS incomplete isoforms are available as a GTF file in Supplementary Data 6.

## Code availability

The code for this study is available on request from the corresponding author.

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

## Acknowledgements

This study was supported in part by a Grant-in-Aid for Scientific Research (B) (18H02849) and Grant-in-Aid for Challenging Research (21K19501) from the MEXT of Japan. This study was also supported by a grant from Medical Research Center Initiative for High Depth Omics. We thank K. Kobayashi for her technical assistance.

## Author contributions

K.Yamaguchi., K.I., H.S., K.F., K.Yamamoto, and Y.K. designed the research project. K. Yamaguchi conducted bioinformatics analysis with the help of K.I and Y.K. K.Yamaguchi, A.S., Y.T., H.T., S.S., Y.N., and H.S. performed short-read RNA-sequencing. F.M. and T.T. contributed sam-ples and data for long-read RNA-sequencing. K.Yamaguchi wrote the manuscript with critical input from K.I. and Y.K.

## Competing interests

The authors declare no competing interests.
