## [Peer Review File · Nature Communications]

Splicing QTL analysis focusing on coding sequences reveals pathogenicity of disease susceptibility loci.REVIEWER COMMENTS

Reviewer #1 (Remarks to the Author):

The authors re-analyse a cell sorted blood/PBMC eQTL cohort they previously used to map eQTLs and sQTLs. Their main innovation is to group isoforms into those corresponding to the same coding sequence (CDS). They find this somewhat increases power to detect isoform ratio QTLs, presumably because it reduces the uncertainty in read assignment between the merged isoforms, which will by construction have highly similar sequences. Additionally, by focusing on these changes, that will change the protein sequence (and therefore potentially structure and/or function), there is the potential to find more likely disease/trait-impacting changes. The authors themselves acknowledge however that changes to the 5' or 3' UTR could also change for example translation efficiency or stability respectively. Some CDS are "incomplete" in that part of the protein sequence is unknown. To help resolve some of these events the authors perform targeted long read sequencing and analysis.

The paper is quite clearly written, although I think the idea of merging isoforms with the same CDS could be expressed a little more clearly in the text. Maybe explicitly saying that the TPM values are added would be helpful rather than using the term "integration". The figures do significantly help with the explanation however.

Significance. As contributions to the field the work does feel somewhat incremental: the LRS data generated is modest in scale, the idea of merging same-CDS isoforms is simple and straightforward. However, I have not seen it done and it does make for an interesting approach to sQTL analysis in between normal irQTL and junction based QTL mapping. Plus the individual gene examples presented are well presented and convincingly.

I do have some minor technical comments/concerns.

1. Defining cell type specificity of QTLs by the number of cell types where significance is observed is statistically fraught because there may be difference in power across cell types due to difference in data quality or expression variability (not to mention the difference between a gene just not being expressed vs the SNP no longer having an effect). GTEx has approaches (MASHR etc) to handle this appropriately. (I'm actually not too worried about this in this setting since the different cell type data is coming from the same cohort, but the statistics should be done properly anyway).

2. trans-eQTLs at this sample size (~100) will be severely underpowered (unless you're detecting artifacts of course). The authors work around this using GSEA, but this is somewhat unsatisfying. For at least some of the cell types there are larger publicly datasets they could try to leverage.

Finally, I disagree somewhat with the statement that junction-based sQTL approaches have underestimated cell-type specificity of sQTLs because they altTSS QTLs. By most common definitions, altTSS QTLs are not sQTLs, even if they might sometimes be (incorrectly) detected as such. In particular, they involve pre, rather than post, transcriptional regulation.

Overall I think this is a solid piece of work. I suggest acceptance following minor revisions.

Reviewer #2 (Remarks to the Author):

This manuscript reports a study of splicing QTL focusing on coding sequences. Specifically, the authors developed two strategies, one is to integrate isoforms with the same coding sequence and analyze them collectively; the other is to examine incomplete CDS isoforms and focus on their isoform QTL patterns. The authors carried out long-read capture RNA-seq to confirm a small number of incomplete CDSs. The manuscript highlighted a few genes harboring sQTL and their potential disease-relevance.

The most notable aspect of the work is the integration approach where isoforms with the same CDS are integrated in sQTL analysis to hopefully reach more power and focus on analysis per protein isoform. However, the overall work is quite limited in terms of methodologies, validations and the potential impact of their findings.

In their approach, the authors first estimate isoform expression in RNA-seq data using Cufflinks. It is well known, as pointed out in the manuscript, that isoform expression is not reliably estimated from short read RNA-seq in general. Since the approaches used in this work closely rely on such estimates, this topic needs a more careful treatment. Comparison of the results from multiple methods (in addition to Cufflinks) is necessary. Evaluation and validation of the accuracy of such estimates are also needed.

The authors did not evaluate the accuracy of the predicted i-rQTLs, or other predictions. What are the false positive rate? What is the sensitivity and specificity of their approach?

The manuscript went into great depth on a small number of genes, by describing the QTL patterns, the GWAS SNPs and their potential disease relevance. However, such descriptions are largely speculations. There is little experimental support on these claims. Thus, although such findings are potentially interesting, they are not yet compelling scientifically. Experimental validations are needed to support their predictions.

Some over-interpretation of their results exist. For example, for Fig. 2a, the authors described in the text (lines 84-85) "I-rQTL with high cell-type specificity ... had a large peak near the TSS." What is defined as "large". A visual inspection of Fig 2a does not show clear difference. There is no confidence intervals of statistics for this figure. Thus, the interpretations are highly arbitrary.

The value of incomplete CDS isoforms is unclear. They appear to be lowly expressed. Out of the >28K such isoforms, only 175-337 had I-rQTL effects. Considering the unknown rate of false positives, it is questionable whether such a small number of i-rQTL genes are functionally relevant. The authors only attempted to confirm the existence of 37 of them using long read data, which were also predicted by LeafCutter analysis, which again bring the question how effective their original I-rQTL prediction methods are.

The trans-eQTL effect of sQTL appears to be an interesting topic to pursue. However, the analysis presented here is quite superficial, far from a comprehensive treatment. Rather than presenting global analysis results, the authors focused on an example gene. It is hard to evaluate whether trans-eQTL effect of sQTL is a widespread phenomenon with biological significance or not.

Dear Editors and Reviewers

Thank you very much for reviewing our manuscript and offering valuable advice.

We have addressed your comments with point-by-point responses and revised the manuscript accordingly.

Responses to the Comments by Reviewer #1:

1. The paper is quite clearly written, although I think the idea of merging isoforms with the same CDS could be expressed a little more clearly in the text. Maybe explicitly saying that the TPM values are added would be helpful rather than using the term "integration". The figures do significantly help with the explanation however.

Reply:

Thank you for this helpful comment. We agree that it was difficult for readers to understand the meaning of "integration", and we would like to adopt your phrase "the TPM values are added", in the first appearance of "integration" as below. We retained the term "integration" which we think represents a core concept throughout the paper.

>>>

To focus on changes in protein sequences caused by sQTL effects, we integrated isoforms with the same CDS by adding their expression levels (FPKM) and then re-analyzed i-eQTL/i-rQTL using these integrated-isoforms (i²-eQTL analysis, integrated-isoform eQTL analysis; i²-rQTL analysis, integrated-isoform ratio QTL analysis).

(page 4, line 115)

2. Defining cell type specificity of QTLs by the number of cell types where significance is observed is statistically fraught because there may be difference in power across cell types due to difference in data quality or expression variability (not to mention the difference between a gene just not being expressed vs the SNP no longer having an effect). GTEx has approaches (MASHR etc) to handle this appropriately. (I'm actually not too worried about this in this setting since the different cell type data is coming from the same cohort, but the statistics should be done properly anyway).

Reply:

As you pointed, there may be differences in power across cell types. We tried to minimize the difference in data quality by using the datasets obtained from one laboratory with the same methods, including approximately the same sample size and the read depths between the cell types. For the expression variability, we only analyzed commonly expressed genes in all subsets, although the quantitative (but not qualitative) variations of expression levels, especially those for isoforms, might have affected the results.

We took your comment seriously and performed additional analysis using mash R. Because this analysis has more statistical power than our analysis of single cell-subset, we identified far more i-rQTL genes compared to our previous analysis (9,094 genes vs 2,494 genes for five subsets) but less single cell-type specific i-rQTL genes (13 genes for five subsets). However, the distribution showed the same trend along with

the cell specificity (= the number of cell types that shared the i-rQTL effects), confirming our original results (Supplementary Figure 5). We used the original results for the main text to keep a consistency with the downstream analyses.

Supplementary Figure 5. Differential distribution of i-rQTL lead variants by cell type specificity using mash analysis

The results of i-rQTL analysis were meta-analyzed using mash (multivariate adaptive shrinkage) analysis to assess the cell-type specificity. As a result, 8735 of the 9094 i-rQTL genes had a common i-rQTL effect in all cell types, and only 13 i-rQTL genes had one cell-type specific i-rQTL effect. The distribution of lead i-rQTL variants relative to the gene body for each cell-type specificity were shown. The cell-type specificities were classified as in 1-2 cell type(s), 3-4 cell types, and 5 cell types to avoid extreme differences in the number of genes in each category.

3. trans-eQTLs at this sample size (~100) will be severely underpowered (unless you're detecting artifacts of course). The authors work around this using GSEA, but this is somewhat unsatisfying. For at least some of the cell types there are larger publicly datasets the they could try to leverage.

Reply:

We completely agree that the sample size is too small to perform trans-eQTL analysis with regards to both sensitivity and specificity. As you suggested, we performed the same analysis using a larger public data, GEUVADIS samples (n=373), and obtained the same results (Supplementary Figure 8, please see the next page). Therefore, we believe our approach to perform trans-eQTL analysis in combination with GSEA analysis would yield useful information for the functional consequences of sQTL.

Supplementary Figure 8. GSEA analysis for i-rQTL isoforms of SNRPC using GEUVADIS RNA-seq dataset

We performed the same GSEA analysis as in Figure 6g for validation, using RNA-seq data from 373 LCLs (European population) of the GEUVADIS project. Similar to the results using peripheral blood in the main text, SNRPC isoforms affected the interferon signature genes. (SNRPC-201 NES 2.07, q-value < 0.0001; SNRPC-203 NES -2.02, q-value < 0.0001).

4. Finally, I disagree somewhat with the statement that junction-based sQTL approaches have underestimated cell-type specificity of sQTLs because they altTSS QTLs. By most common definitions, altTSS QTLs are not sQTLs, even if they might sometimes be (incorrectly) detected as such. In particular, they involve pre, rather than post, transcriptional regulation.

As you pointed out, altTSS QTLs are not sQTLs in a narrow sense of sQTLs that really involve splicing events. However, the major tools used for sQTL analysis like LeafCutter do not precisely distinguish altTSS QTL signals from sQTL signals in a narrow sense. Therefore, the term sQTL has been used ambiguously. In the present study, we redefined all of the QTLs that change isoform ratio as “sQTLs in a broad sense” so that we can focus on changes in isoforms with different protein structures. To avoid misinterpretation by readers, we have clarified the sentence as follows;

“This analysis can identify the full-length of sQTL isoforms, which is important for assessing functional changes in proteins. In addition, this analysis can also detect sQTL effects in a broad sense including those caused by alternative TSS or alternative TES (alternative PAS usage) without changes in junctions.” (page 7, line 241)

Another issue raised by your comment is whether junction-based sQTL approaches have really underestimated cell-type specificity of sQTLs because they cannot sufficiently detect altTSS QTLs. For the response to comment of another reviewer, we analyzed the sensitivity of our analysis using a simulation

RNA-seq data set. As a results, the sensitivity of i-rQTL analysis for altTSS QTL was superior to LeafCutter (Supplementary Fig 2a), although LeafCutter could also detect some of altTSS QTL.

If there exists only two altTSS isoforms that share all the junctions, it is impossible for LeafCutter to distinguish these isoforms. This is not only the case for altTSS isoforms but also for the alternative PAS isoforms. We showed examples for these in Supplementary Figure 11a and 11b, respectively, whose sQTL effects were detected by i-rQTL analysis but not in LeafCutter using the simulated data. Of such differentially expressed altTSS/PAS isoforms (21 genes among 200 genes in the simulated data), 12 were significantly detected by i-rQTL analysis but only one gene was detected in LeafCutter. The only gene detected by LeafCutter had another isoform with a mutually exclusive junction, which allowed junction clustering and detection of sQTL effect by LeafCutter (Supplementary Figure 11c, please see the next page).

We think our additional analysis using the simulated data confirmed that LeafCutter had a difficulty in detecting sQTL effects with alternative TSS and might have underestimated the cell type specificity, we left the description as it was.

Supplementary Figure 2. Accuracy of i-rQTL analysis evaluated using simulation data

- a. Sensitivity and specificity of i-rQTL analysis or LeafCutter or a combination of both analyses for each effect size.

Supplementary Figure 11. Examples of isoforms differentially detected by i-rQTL analysis and LeafCutter.

a. SNRPE gene had two isoforms with different TSS, and the long isoform had all the junctions of the short isoform. The sQTL effect of this gene was identified by i-rQTL analysis but not by LeafCutter.

b. GDAP2 gene had two isoforms with different PAS, and the long isoform had all the junctions of the short isoform. The sQTL effect of this gene was identified by i-rQTL analysis but not by LeafCutter.

c. SRSF11 gene had three isoforms. The bottom isoform had an alternative TSS, and all its junctions were shared by the other two isoforms. The latter two isoforms had a mutually exclusive junction, which is focused in the box. The sQTL effect of this gene was identified by both i-rQTL analysis and LeafCutter.

Responses to the Comments by Reviewer #2:

1. In their approach, the authors first estimate isoform expression in RNA-seq data using Cufflinks. It is well known, as pointed out in the manuscript, that isoform expression is not reliably estimated from short read RNA-seq in general. Since the approaches used in this work closely rely on such estimates, this topic needs a more careful treatment. Comparison of the results from multiple methods (in addition to Cufflinks) is necessary. Evaluation and validation of the accuracy of such estimates are also needed.

Reply:

Thank you for this important comment. We admit that we have not fully evaluated the accuracy of the methods, especially for isoform ratio quantification.

First, according to the previous reports on quantification of isoform expression (not isoform ratio), Cufflinks, salmon, and RSEM showed comparable accuracy (BMC Bioinformatics. 2021 May 25;22(1):266., BMC Genomics

. 2017 Aug 7;18(1):583.). We also evaluated the accuracy of quantification for both isoform and isoform ratio using a simulation RNA-seq dataset created by polyester (Bioinformatics. 2015 Sep 1; 31(17): 2778–2784.). We simulated the isoform expression of genes located in chr1 with an actual average expression of FPKM \geq 0.5 in 105 monocyte samples. Average expression levels of each gene, read length, and the number of reads were set as the same as those in the actual data (126bp paired-end and 4 million reads derived from chr 1).

We obtained isoform FPKM and ratios using Cufflinks, salmon, and RSEM and calculated the Spearman correlation coefficient between them and the true isoform FPKM and ratios. As in the previous report above, the correlation coefficient for isoform FPKM were comparable between the methods (0.940-0.958) (Supplementary Figure 1a, please see the next page). In contrast and unexpectedly, the correlation coefficient for isoform ratio were very high for all three methods (\sim 0.99) (Supplementary Figure 1b, please see the next page). This suggests that the same biases, which occurred in the quantification of individual gene loci, might be canceled out, when calculating the ratio of isoforms. In a real RNA-seq data, quantification of isoform ratio may be more inaccurate because the gene models are more complex and incomplete. Therefore, we think it important to confirm the gene models, as we did for the CDSI incomplete isoforms by capture long-read RNA sequencing.

Because salmon showed the largest decrease in accuracy in the absence of annotation of highly expressed isoforms. (BMC Bioinformatics. 2021 May 25;22(1):266.), we left to choose Cufflinks, which showed the second accuracy for the isoform ratio calculation ($r=0.9894$) next to salmon ($r=0.9898$).

Supplementary Figure 1. Accuracy of three isoform quantification methods evaluated using RNA-seq simulation data

a. Mean of Spearman correlation coefficients between isoform FPKMs calculated using the three methods and the true isoform FPKMs in simulation data. The whiskers in the figure indicate 95% confidence intervals. The respective means are Cufflinks 0.9410, salmon 0.9530, and RSEM 0.9584.

b. Mean of Spearman correlation coefficients between isoform ratios calculated using the three methods and the true isoform ratios in simulation data. The whiskers in the figure indicate 95% confidence intervals. The respective means are Cufflinks 0.9894, salmon 0.9898, and RSEM 0.9889.

2. The authors did not evaluate the accuracy of the predicted i-rQTLs, or other predictions. What are the false positive rate? What is the sensitivity and specificity of their approach?

Reply:

We appreciate this another important comment. Although i-rQTL analysis (trQTL analysis) has long been used in various studies (Nature. 2013 Sep 26;501(7468):506-11.), surprisingly, its accuracy has not been fully evaluated. Therefore, we evaluated the accuracy of i-rQTL analysis using simulation data as described in the previous comment.

We randomly selected 200 genes (approximately 20% of the expressed genes in chr 1), in which two or more isoforms were expressed. Then, we simulated the data as one of isoforms was differentially expressed isoform. Next, we randomly selected one of common variants (MAF ≥ 0.05) in each gene region as a simulated lead QTL variant for the isoform expression. The QTL effect size was set to three patterns of fold changes for each alternative allele: 1.2, 1.5, and 2.0.

As a result, the sensitivity/specificity of the simulated i-rQTL analysis were 91.0%/97.8% and 97.0%/96.6%, for the fold change of 1.5 and 2.0 respectively. The sensitivity i-rQTL analysis for the fold change of 1.2 was low (52.5%), but the specificity remained high (98.7%) (Supplementary Figure 2a, please see the next page). Because more than half of the effect sizes of isoform eQTLs in real data had allelic fold changes 1.2 or higher (Supplementary Figure 2d, please see the next page), the false positive rate (=100% - specificity) of our i-rQTL analysis might be unexpectedly low (less than 5%). This may reflect the unexpectedly high accuracy of

isoform ratio estimation, as described in the previous comment.

Of note, the specificity of the sQTL analysis using LeafCutter was very high, but contrary to our expectations, the sensitivity was very low. The reason for this was that 40% of the genes were excluded in the analysis due to insufficient clustering of the junction. However, as the advantages of LeafCutter were highlighted in realistic data, rather than in simulated data (Nat Genet. 2018 Jan;50(1):151-158.), the sensitivity of LeafCutter may be higher in realistic data.

Supplementary Figure 2. Accuracy of i-rQTL analysis evaluated using simulation data

- a. Sensitivity and specificity of i-rQTL analysis or LeafCutter or a combination of both analyses for each effect size.
- d. Effect sizes of isoform eQTL obtained from analysis using real data. The red line indicates the cumulative percentage (right axis).

3. The manuscript went into great depth on a small number of genes, by describing the QTL patterns, the GWAS SNPs and their potential disease relevance. However, such descriptions are largely speculations. There is little experimental support on these claims. Thus, although such findings are potentially interesting, they are not yet compelling scientifically. Experimental validations are needed to support their predictions.

Reply:

We understood the importance of experimental proof of our analysis results. Indeed, we have already attempted to analyze functional consequences of protein sequence changes in several QTLs, but we still have some methodological challenges. We have conducted isoform specific knock-down experiments using siRNA in LCL and THP-1 cells, but the knock-down efficiency was insufficient (<50%) to obtain convincing results. There are several reasons for this; 1) The sequence length to design the siRNA are very limited (only the unique region of isoform, such as sequences at alternative junction, can be used), making it difficult to design good siRNAs; 2) majority of isoforms we identified were functioning in immune cells, which had low transfection efficiency by current transfection protocols (AMAXA, Lipofectamine, etc).

Gene knock-down using CRISPR-Cas9 would be more straightforward, but the same challenges (limited target sequences and transfection efficiency) would occur. Moreover, genome editing at DNA level would influence not only the targeted isoform but also other splicing events. As we think technical breakthroughs may be needed to comprehensively analyze the function of disease-relevant isoforms, we would like to leave this important issue to our future study or other scientists specializing in molecular biology.

4. Some over-interpretation of their results exist. For example, for Fig. 2a, the authors described in the text (lines 84-85) “i-rQTL with high cell-type specificity ... had a large peak near the TSS.” What is defined as “large”. A visual inspection of Fig 2a does not show clear difference. There is no confidence intervals of statistics for this figure. Thus, the interpretations are highly arbitrary.

Reply:

Thank you for this comment. Indeed, many of our descriptions were arbitrary rather than statistics-based. First, we statistically re-evaluated the distribution of the lead i-rQTL variants in Figure 1e. The distribution of the i-rQTL was initially described as having two peaks in the promoter region and the 3'-UTR region. When we tested for multimodality using the Silverman test, we found that the distribution of i-rQTL variants was trimodal, with another peak in the gene body (Supplementary Figure 4, please see the next page). The presence of the i-rQTL peak in the Gene body may be plausible because it may reflect the variants at canonical splice sites. We believe that the statistical evaluation allowed us to interpret the distribution more correctly. Next, we added 95% confidence intervals calculated by pairwise bootstrapping to Figure 2a (please see the next page). This confidence interval clarifies that there are significant differences in the peaks near TSS for cell type specificity.

We added a sentence in the main text as follows;

“(A trimodal distribution with two additional peaks in the gene body and TES was detected by Silverman test, $p = 0.039$).” (Page 4)

Supplementary Figure 4. Evaluation of multimodality of lead i-rQTL distribution using silverman test

a. p-values of the silverman test in each mode performed on the distribution of i-rQTL variants in Figure 1e.

b. The position and bandwidth of the peaks for each mode. On the x-axis, TSS corresponds to -15,000 and TES to 15,000.

Figure 2. Cell type specific i-rQTL

a. For each cell-type specificity, distributions of lead i-rQTL variants to the gene body are shown. The shade shows 95% confidence intervals for the distribution of 1 cell-type specific i-rQTLs.

5. The value of incomplete CDS isoforms is unclear. They appear to be lowly expressed. Out of the >28K such isoforms, only 175-337 had i-rQTL effects. Considering the unknown rate of false positives, it is questionable whether such a small number of i-rQTL genes are functionally relevant. The authors only attempted to confirm the existence of 37 of them using long read data, which were also predicted by LeafCutter analysis, which again bring the question how effective their original i-rQTL prediction methods are.

Reply:

We thank the reviewer for this essential comment. We selected the incomplete CDS isoforms from the Gencode annotations, because it represented the isoforms that were understudied owing to its low expression but having a potential to change the gene function qualitatively. As you apprehended, both isoform annotation itself and QTL analysis on these CDSI isoforms would comprise false positive findings. Therefore, we adopted a conservative criterion, at the cost of comprehensiveness, for the determination of CDSI i-rQTL using the results of both i-rQTL analysis and LeafCutter. Furthermore, as mentioned to the comment 2, i-rQTL analysis using a simulated dataset showed its high specificity, and we think the majority of 37 loci we examined are true positive.

We admit that LeafCutter could also predict the presence of CDSI isoforms and sQTL effects on these isoforms, but confirmation of full sequences by long-read capture RNA-seq would sometimes yields additional information as was clearly presented by the case of ATXN2L, in which splicing events occurred in the CDSI specific junction influenced the alternative usage of TSS (though the cause-and-result relationship might be opposite in this locus).

With the advent of long-read sequencing, a huge variety of isoforms has been apparent in past few years. We admit our analysis lacked comprehensiveness, but we believe our approach to select isoforms with low expression but with potential functional relevance would dissect unknown etiology of complex traits.

6. The trans-eQTL effect of sQTL appears to be an interesting topic to pursue. However, the analysis presented here is quite superficial, far from a comprehensive treatment. Rather than presenting global analysis results, the authors focused on an example gene. It is hard to evaluate whether trans-eQTL effect of sQTL is a widespread phenomenon with biological significance or not.

Reply:

We agree that our trans-eQTL analysis is very superficial from the viewpoint of comprehensiveness. Because the sample size (~100) precluded us from a genome-wide survey that incurred a multiple hypothesis burden, we decided to choose a representative locus for this analysis. The SNRPC locus (Fig 6) was one of good candidates, because it contained multiple eQTL/sQTL effects on multiple genes but the responsible gene/QTL effect had not been determined yet. In addition, to complement our trans-eQTL analysis, we combined GSEA analysis which made it possible to evaluate moderate trans-eQTL effects of a single locus as a whole.

As a similar issue was raised by the other reviewer which questioned the robustness of our findings, we performed the same analysis on the SNRPC locus using a larger GEUVADIS dataset (n=373) and obtained the same results (Supplementary Figure 8). In addition, we evaluated the trans-eQTL effects of sQTL with protein-structure changes using the eQTLgen database (n = 31,684). We examined whether our sQTL variants (or proxy variants with $r^2 \geq 0.8$) have trans-eQTL effect(s) on at least one gene (FDR ≤ 0.05). In comparison with i-rQTL without protein-structure changes, i²-rQTLs with protein-structure changes had a significantly higher proportion of trans-eQTL effects, as expected (44.1 % versus 33.8 %, p = 0.037 by Fisher's exact test, Supplementary Figure 9, please see the next page). This indicates that sQTLs with protein-structure changes explain an important part of variation in the genomic function, and we believe that our approach, trans-eQTL analysis combined with GSEA analysis, is useful for the functional dissection of individual sQTL.

Supplementary Figure 8. GSEA analysis for i-rQTL isoforms of SNRPC using GEUVADIS RNA-seq dataset

We performed the same GSEA analysis as in Figure 6g for validation, using RNA-seq data from 373 LCLs (European population) of the GEUVADIS project. Similar to the results using peripheral blood in the main text, SNRPC isoforms affected the interferon signature genes. (SNRPC-201 NES 2.07, q-value < 0.0001; SNRPC-203 NES -2.02, q-value < 0.0001).

Supplementary Figure 9. Trans-eQTL effect of sQTLs with/without protein-structure changes

Trans-eQTL effects of sQTLs with protein-structure changes were evaluated using eQTLgen database. In comparison with i-rQTL without protein-structure changes (142 variants), i^2 -rQTLs with protein-structure changes (188 variants) had a significantly higher proportion of trans-eQTL effects, as expected (44.1 % versus 33.8 %, $p = 0.0366$ in one-sided Fisher's exact test).

REVIEWERS' COMMENTS

Reviewer #1 (Remarks to the Author):

The authors have done a thorough job addressing both my concerns and those of the other reviewer. I would recommend publication barring one small technical issue: I thought they were adding TPMs for the isoforms sharing their CDS, but seemingly they are adding FPKMs, which IMO is less interpretable. I would like them to check that it doesn't make too much difference doing this (I don't need to see a revision though).

Reviewer #2 (Remarks to the Author):

NA

Dear Editors and Reviewers

Thank you very much for taking the time to review the manuscript again.

Responses to the Comments by Reviewer #1:

1. I thought they were adding TPMs for the isoforms sharing their CDS, but seemingly they are adding FPKMs, which IMO is less interpretable. I would like them to check that it doesn't make too much difference doing this (I don't need to see a revision though).

Reply:

This comment may mean that TPM, rather than FPKM, is often used these days for gene expression comparisons such as DEG analyses. However, we used FPKM because Cufflinks that we used in the present study can only output FPKM but not TPM.

In i-rQTL and i²-rQTL analyses, which feature the present study, we used isoform ratio instead of gene expression. Because the isoform ratio, whether calculated using FPKM or TPM, shows exactly the same value (please note the conversion formula between TPM and FPKM below), the results of i-rQTL and i²-rQTL analyses would not make difference.

$$TPM_i = \left(\frac{FPKM_i}{\sum_j FPKM_j} \right) \cdot 10^6 \quad (i \text{ is for the isoform of interest, } j \text{ is for all the isoforms of entire genome}).$$